# Enhancement of Thermophilic Biogas Production from Palm Oil Mill Effluent by pH Adjustment and Effluent Recycling

Apinya Singkhala [1], Chonticha Mamimin [2], Alissara Reungsang [2,3] and Sompong O-Thong [1,4,*]

1. Biotechnology Program, Department of Biology, Faculty of Science, Thaksin University, Phatthalung 93210, Thailand; s25.apinya@gmail.com
2. Department of Biotechnology, Faculty of Technology, Khon Kaen University, Khon Kaen 40002, Thailand; chonticha51@gmail.com (C.M.); alissara@kku.ac.th (A.R.)
3. Research Group for Development of Microbial Hydrogen Production Process from Biomass, Khon Kaen University, Khon Kaen 40002, Thailand
4. International College, Thaksin University, Songkhla 90000, Thailand
* Correspondence: sompong@tsu.ac.th; Tel.: +66-94-914-9923

**Abstract:** A sudden pH drops always inhibits the anaerobic digestion (AD) reactor for biogas production from palm oil mill effluent (POME). The pH adjustment of POME by oil palm ash addition and the biogas effluent recycling effect on the preventing of pH drop and change of the archaea community was investigated. The pH adjustment of POME to 7.5 increased the methane yield two times more than raw POME (pH 4.3). The optimal dose for pH adjustment by oil palm ash addition was 5% $w/v$ with a methane yield of 440 mL-CH$_4$/gVS. The optimal dose for pH adjustment by biogas effluent recycling was 20% $v/v$ with a methane yield of 351 mL-CH$_4$/gVS. Methane production from POME in a continuous reactor with pH adjustment by 5% $w/v$ oil palm ash and 20% $v/v$ biogas effluent recycling was 19.1 ± 0.25 and 13.8 ± 0.3 m$^3$ CH$_4$/m$^3$-POME, respectively. The pH adjustment by oil palm ash enhanced methane production for the long-term operation with the stability of pH, alkalinity, and archaea community. Oil palm ash increased the number of *Methanosarcina mazei* and *Methanothermobacter defluvii*. Oil palm ash is a cost-effective alkali material as a source of buffer and trace metals for preventing the pH drop and the increased methanogen population in the AD process.

**Keywords:** biogas production; biogas effluent recycling; low pH inhibition; oil palm ash; palm oil mill effluent; thermophilic condition

## 1. Introduction

Palm oil mill effluent (POME) is the main wastewater generated from palm oil extraction plants, which has a low pH (4.9), high chemical oxygen demand (COD) (68 g/L), and high oil content (6.2 g/L) [1]. POME is commonly treated using the anaerobic digestion (AD) process with pollution reduction and energy recovery in sustainable ways with high commercial applications in terms of biogas [2]. The AD of POME is usually operated at high hydraulic retention time (HRT) and low organic loading rate (OLR) to prevent pH drops and fast acidification leading to the requirement of a large reactor volume [3]. The 0.5 m$^3$ of POME is generated from 1 tonne of fresh fruit bunches (FFB) [4]. It is estimated that 400–600 m$^3$/d is generated from a mill plant with a processing capacity of 800–1000 tonne/d. The current commercial biogas reactors size of 6000–10,000 m$^3$ are constructed at most of the oil palm extraction plants to save the construction cost and land used, leading to a low HRT of 10–25 days for reactor operation, corresponding to a POME flow rate of 400–600 m$^3$/d. The characteristics and quantity of POME vary depending on FFB loading to the extraction process, FFB quality, and cropping season. The COD content of POME in the high harvesting season is 20–50% higher than in the low harvesting season [5]. The variation of feedstock composition and high organic loading operation

of readily biodegradable POME can easily drop the pH in the AD reactor resulting in a low pH inhibition [1]. Therefore, pH adjustment of POME before feeding to the AD reactor is needed to keep the pH inside the AD reactor at an optimal range of 6.5–7.5 and favor the growth of methanogenic archaea. Adjustment of the pH of the AD reactor to near-neutral is often applied to enhance the buffering capacity of the AD system against VFA accumulation, with low cost and smooth operation [6].

Alkaline addition to the AD reactor improves the buffering capacity to meet the requirements of the microbial populations while balancing the growth of acidogenic bacteria and methanogenic archaea [7]. The large volume of POME requires a large amount of alkaline chemicals for long-term pH adjustment. Cheap materials such as ash and biogas effluent are used to adjust the pH in AD reactors [8]. Ash has been found to improve acid-neutral capacities and increase gas production of municipal solid waste [9]. The oil palm biomass is often used as boiler fuel by palm oil mill plants to produce steam for electricity generation and palm oil extraction, generating many oil palm ash [10]. The addition of oil palm ash at 8.0% $w/v$ can recover the inhibited sludge from a full-scale biogas reactor with increased pH and buffer capacity [11]. The addition of ash can act as a pH adjustment in the AD reactor by enhancing the microbial growth rate by releasing alkali and trace metals [12]. Ash also contains trace metals (Co and Fe) that were crucial for the activity of enzymes in methanogenic archaea and increased the number of archaea [13]. Recycling 50% ($v/v$) of biogas effluent into the AD reactor for pH adjustment increased methane yield by up to 21% under mesophilic conditions [14]. The biogas effluent could increase the buffer capacity and supply nitrogen sources to the anaerobic microorganisms [15]. Lu et al. [16] reported that the recycled biogas effluent could buffer the acidified AD process. The recirculation of biogas effluent is helpful for the renovation of methane-producing microorganisms and maintains optimal pH [17]. Feng et al. [18] observed that the recirculation of biogas effluent to the AD reactor could alleviate pH reduction with increased methane production by 156%.

The POME had high organic content in terms of COD and high temperature (80–92 °C). Thermophilic AD is suitable for POME due to low cooling cost requirements and increases in the rates of chemical and enzymatic reactions, increasing thermodynamic favorability of methane production reactions [19]. The low pH condition in the AD process accelerates the decay of methanogenic archaea resulting in complete failure of the AD reactor. The specific decay rate of methanogenic archaea is increased 10 times in a low pH environment (pH 5.1) compared to neutral pH (pH 7.0). The methanogenic activity was included in the IWA Anaerobic Digestion Model no.1 (ADM1) to predict the times required to recover methanogens from acidic failure in biogas plants [20]. The number of methanogenic archaea was decreased by 71–79% in a low pH environment (pH 5.1) comparing to neutral pH (pH 7.0), while a low pH environment reduces the number of bacteria by 25% [20]. The imbalance in the growth and number between acidogenic bacteria and methanogenic archaea occurred in a low pH environment. Ash provides buffer capacity and tract metals to prevent sudden pH drops in the AD reactor, while biogas effluent provides buffer capacity and nitrogen source. However, the effect of buffer capacity and tract metals from oil palm ash on the growth and number of acidogenic bacteria and methanogenic archaea is lacking and there are no reports on buffer capacity and nitrogen sources from biogas effluent on the growth and number of acidogenic bacteria and methanogenic archaea.

This work investigated the proper amount of oil palm ash addition and biogas effluent recycling for adjusting the pH of POME before feeding into the AD reactors without disturbing the microbial communities and preventing the sudden pH drop. A continuous reactor was used to evaluate the long-term effect of oil palm ash addition and biogas effluent recycling on the stability of pH and alkalinity with the change of bacteria community and archaea community.

## 2. Materials and Methods

### 2.1. Raw Materials

Palm oil mill effluent (POME), oil palm ash, biogas effluent, and methane-producing sludge were obtained from the Southern Palm Oil (2521) company limited, located in Surat Thani Province, Thailand. POME was generated from a vertical clarifier tank at 95 °C. POME was collected from an equalization tank through an oil palm extraction process with a production capacity of 800 tonnes FFB/d. Biogas effluent was collected from the effluent of a methanogenic tank. POME and biogas effluent were kept at 4 °C until use. Before use, POME and biogas effluent were incubated in a water bath at 55 °C for 2 h. The oil palm ash was collected from biomass-fired boilers containing both bottom ash and fly ash. The compositions of POME, oil palm ash, and biogas effluent are shown in Table 1. The metals composition in oil palm biomass ash was analyzed using an X-ray fluorescence spectrometer, as described by Tan et al. [21]. The methane-producing sludge was collected from the biogas reactor operated under thermophilic conditions (55 °C) using POME as the substrate. The sludge was incubated for one week to reduce the remaining substrate. The sludge with volatile suspended solids (VSS) content of $15.0 \pm 1.2$ g/L was used as an inoculum. POME and biogas effluent was measured for alkalinity, pH, total volatile fatty acids, COD, total nitrogen (TN), total solids (TS), volatile solids (VS), and oil and grease. Oil palm ash was measured for total solids (TS), pH, and alkalinity.

**Table 1.** Characteristics of POME, biogas effluent, and oil palm ash.

| Characteristics | POME | Oil Palm Ash | Biogas Effluent |
|---|---|---|---|
| Total solids (g/L) | 67 | 86 * | 13.4 |
| Volatile solids (g/L) | 58 | - | 4.8 |
| Chemical oxygen demand (g/L) | 97 | - | 7.5 |
| Total nitrogen (g/L) | 2.5 | - | 0.2 |
| Oil and grease (g/L) | 12.4 | - | 1.1 |
| Volatile fatty acids (g/L) | 3.3 | - | 0.12 |
| Alkalinity (g-CaCO$_3$/L) | 0.15 | 11.3 | 5.9 |
| pH | 4.3 | 11.2 | 8.8 |

* Total solids of ash expressed as g/kg.

### 2.2. Batch Experiments

Methane production from raw POME (pH 4.3), pH adjustment to 7.5 by NaHCO$_3$, and biogas effluent were carried out in a batch reactor, as previously described by Angelidaki et al. [22]. The experiments were performed in 500 mL glass serum bottles with a working volume of 300 mL. Methane production of raw POME was determined at VS loading of 7.5, 13, and 20 g-VS/L without pH adjustment. The methane potential of pH adjustment POME (pH 7.5) was investigated at VS loading of 7.5, 13, and 20 g-VS/L, respectively. Methane production of biogas effluent was investigated at 50%, 70%, and 100%, respectively. Biogas production from pH adjustment POME by oil palm ash was tested at 0, 2.5, 5, 7.5, 10, 12.5, and 15% (*w/v*), respectively, with fixed VS loading of 7.5 g-VS/L. Biogas production from pH adjustment POME by biogas effluent recycling was tested at 5, 10, 15, 20, 25, and 30% (*v/v*), respectively, with fixed VS loading of 7.5 g-VS/L. The substrate to inoculum (S/I) was 2:1 based on VS basis of POME and anaerobically digested sludge. The serum bottles were continuously flushed with the gas mixture 80% (*v/v*) of N$_2$ and 20% (*v/v*) of CO$_2$ to create the anaerobic conditions. The serum bottles were closed tightly with a butyl-rubber septum and aluminum caps. The serum bottles were incubated at thermophilic temperature (55 °C) for 45 days. All experiments were carried out in triplicate. The biogas volume and biogas composition were monitored daily. The samples were taken from each bottle at the initial and the end of the experiment for pH, VFA, alkalinity, and COD analysis.

### 2.3. Continuous Reactor Operation

Continuous biogas production from pH adjustment POME by oil palm ash addition (reactor R1) and biogas effluent recycling (reactor R2) were evaluated in a 1.5 L continuous stirred tank reactor (CSTR) with a working volume of 1.2 L. The optimum dose for pH adjustment POME by oil palm ash addition and biogas effluent recycling of 5.0% *w/v* and 20% *v/v*, respectively, from the batch experiment was selected to evaluate in a continuous reactor. Figure 1 illustrates a schematic diagram of the experimental apparatus used in this study. Reactor R1 was composed of a mixing tank for mixed oil palm ash with POME, a CSTR reactor for biogas production, and an effluent tank. The reactor R2 was composed of a mixing tank for mixed biogas effluent with POME, a CSTR reactor for biogas production, and a settling tank for the solid-liquid separation biogas effluent. Reactor R1 and R2 had controlled temperature at 55 °C and agitation at 100 rpm. Reactor R1 was fed with pH adjustment POME by 5% (*w/v*) oil palm ash addition. R2 reactor was fed with pH adjustment POME by 20% (*w/v*) biogas effluent addition. Both reactors were inoculated with 960 mL (80% *v/v*) of thermophilic methane-producing sludge. The 240 mL of pH adjustment POME was added to the reactor for the final volume of 1.2 L. The reactors were started with a batch operation for one week before being changed to continuous operation. Reactor R1 and R2 were operated at an HRT of 15 days and OLR of 3.8 g-VS/L/d. Biogas volume and pH were recorded daily, while volatile fatty acid concentrations were measured every four days. The $H_2$, $CO_2$, and $CH_4$ in biogas were analyzed daily by gas chromatography. The effluent samples from the reactors were analyzed for VFA, pH, and alkalinity. The sludge samples were taken every week from both reactors for dynamic microbial population analysis.

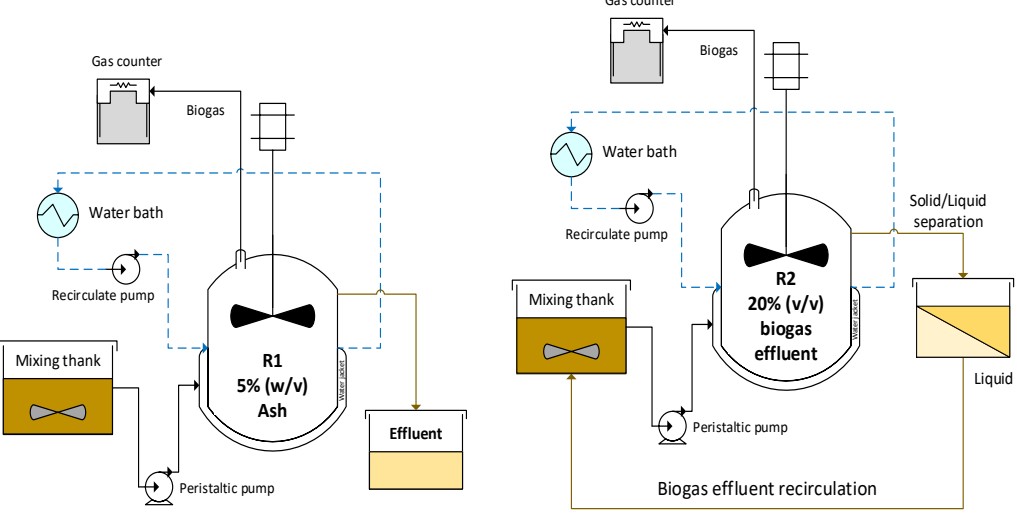

**Figure 1.** Schematic diagram of continuous biogas production from pH adjustment POME by oil palm ash (reactor R1) and biogas effluent recycling (reactor R2).

### 2.4. Analytical Methods and Calculation

The alkalinity, pH, total volatile fatty acids, COD, total nitrogen (TN), total solids (TS), volatile solids (VS), and oil and grease were analyzed by standard methods [23]. The biogas volume was measured using a gas meter [24]. The $H_2$, $CO_2$, and $CH_4$ in biogas were analyzed by gas chromatography connected with a thermal conductivity detector (GC-8A Shimadzu, Japan) and Shin-Carbon ST 100/120 column (Restek) [25]. The injection port, oven, column, and detector were used at temperatures of 120, 40, 40, and 100 °C, respectively. The high purity argon was used at 14 mL per min as the carrier gas. The 0.5 mL of a gas mixture (10% H2, 10% N2, 50% $CH_4$, and 30% $CO_2$; based on % of mol) was used as a standard gas for GC calibration. The liquid samples from the AD reactor were

filtered through a 0.2 μm nylon membrane and acidified by 30% ($v/v$) phosphoric acid to pH 3–3.2 [26]. VFAs were analyzed using a gas chromatograph connected with stabilwax®-DA fused silica column and flame ionization detector (GC-17A Shimadzu, Japan). The column, injector, and detector temperatures were maintained at 85, 230, and 240 °C. The high purity helium was used at a 30 mL/min flow rate as the carrier gas. High purity nitrogen was used at a flow rate of 50 mL/min as the make-up gas, and hydrogen and air zero were used at a flow rate of 50 mL/min as flam gas. Calibration was done with a standard free fatty acids mixture containing 1000 μg/mL. Filtrated liquid samples were acidified to pH 3–3.2 with 30% ($v/v$) phosphoric acid before used for VFAs analysis. As in Equation (1), the first-order kinetic model was used to determine the hydrolysis constants ($k_h$) and determine the biodegradation rate. The $k_h$ was achieved from the slope of the curve according to the protocol of Raposo et al. [27].

$$ln = \frac{B_0}{(B_0 - B_t)} \, k_h t. \tag{1}$$

The cumulative $CH_4$ yield (mL-$CH_4$/gCOD) for time $t$ is $B_t$. The ultimate $CH_4$ yield is $B_0$. The percentage of biodegradability refers to the $CH_4$ yield achieved from an experiment per theoretical $CH_4$ yield calculated from an empirical formula of the substrate according to Bushwell's formula [28]. The methane yield was calculated from the volatile solids of the substrate. The methane production from biogas effluent alone was subtracted from the methane production from pH adjustment by biogas effluent experiment for methane yield calculation. The following equation (Equation (2)) was used to convert methane volume at 55 °C to standard temperature and pressure (STP) conditions (T = 273.15 K, P = 105 Pa).

$$\frac{V_1}{V_2} = \frac{T_1}{T_2} \tag{2}$$

where $V_1$, $V_2$, $T_1$, and $T_2$ were methane volume at 55 °C (L), methane volume at STP (L), incubation temperature (273.15 + 55 K), and temperature of STP (273.15 K), respectively [29].

*2.5. Microbial Community Analysis*

Polymerase chain reaction-denaturing gradient gel electrophoresis (PCR-DGGE) was used to study microbial community structure in the reactor R1 and R2. Sludge samples were collected from bioreactors at days 0, 7, 14, 21, 28, 35, and 42 of operation times. Genomic DNA extractions and PCR-DGGE for bacteria were made as previously described [19]. DGGE profiles were compared using the Quantity One software package (version 4.6.0; Hercules, CA, USA). Most of the bands were excised from the gel and re-amplified. After re-amplification, PCR products were purified and sequenced by Macrogen Inc. (Seoul, Korea). The closest matches for partial 16S rRNA gene sequences were identified by database searches in Gene Bank using BLAST.

**3. Results and Discussion**

*3.1. Biogas Production from Raw POME*

The POME had high organic content in terms of COD (97 g/L) and oil and grease concentration (12.4 g/L). POME also had a high temperature (80–92 °C) and a low pH (4.3) (Table 1). Oil palm ash had a high alkaline property with alkalinity and pH of 11.3 g-$CaCO_3$/L and 11.2, respectively. The chemical composition of oil palm ash is presented in Table 2, which shows high $SiO_2$ (44.84%) and CaO (12.01%), $K_2O$ (4.99%), $P_2O_5$ (4.48%), MgO (3.22%), $Fe_2O_3$ (2.13%), $SO_3$ (1.66%), Cl (1.22), $Al_2O_3$ (1.12%), $MnO_2$ (0.19%), $TiO_2$ (0.14%), CuO (0.07%), SrO (0.04%) and Rb (0.03%). The high alkalinity and high pH of oil palm ash could be beneficial for pH adjusting agents in the AD process of POME. The presence of metals such as Ca, K, Mg, Na, Al, and Fe in the oil palm ash could benefit the AD process by providing alkalinity and trace elements [30]. The addition of ash to the AD process increases macro-and micronutrients (Ca, K, Mg, Si, and P) [31,32]. Biogas

effluent had COD, total nitrogen, alkalinity, pH of 7.5 g/L, 0.2 g/L, 5.9 g/L, and 8.8. Biogas effluent recycling was used to adjust high sulfate wastewater with high methane yield and stable pH in anaerobic baffled reactors [33]. The drawback of biogas effluent recycling by reducing organic loadings due to the wastewater was diluted by the volume of biogas effluent. Oil palm ash can provide buffer capacity and trace metal for AD systems, while biogas effluent can provide buffer capacity and nitrogen source for AD systems. The oil palm ash and biogas effluent could be a suitable and cheap alkaline material for the pH adjustment of POME before feeding to the AD reactor.

**Table 2.** Composition of palm oil biomass ash from a boiler by X-ray fluorescence spectrometer.

| Chemical Constituents | Composition (% Dry Weight) |
| --- | --- |
| Magnesium oxide (MgO) | 3.22 |
| Aluminum oxide ($Al_2O_3$) | 1.12 |
| Silicon dioxide ($SiO_2$) | 44.84 |
| Potassium oxide ($K_2O$) | 4.99 |
| Calcium oxide (CaO) | 12.01 |
| Titanium dioxide ($TiO_2$) | 0.14 |
| Iron oxide ($Fe_2O_3$) | 2.13 |
| Rubidium (Rb) | 0.03 |
| Strontium oxide (SrO) | 0.04 |
| Chlorine (Cl) | 1.22 |
| Manganese dioxide ($MnO_2$) | 0.19 |
| Phosphorus pentoxide ($P_2O_5$) | 4.48 |
| Sulfur trioxide ($SO_3$) | 1.66 |
| Copper oxide (CuO) | 0.07 |

Thermophilic AD is suitable for POME due to low cooling cost requirements and increases in the rates of chemical and enzymatic reactions, increasing thermodynamic favorability of methane production reactions [31]. The $CH_4$ yields of raw POME at initial VS loading of 7.5, 13, and 20 g-VS/L were 236, 228, and 125 mL-$CH_4$/gVS, respectively (Table 3). The raw POME has a maximum $CH_4$ yield of 236 mL-$CH_4$/gVS, corresponding to $CH_4$ production of 13.7 $m^3$-$CH_4$/$m^3$-POME or approximately 22.8 $m^3$-biogas/$m^3$-POME. Methane yield was decreased when increasing VS loading. The low pH property of POME potentially inhibited the AD process. The low pH condition in the AD system has accelerated the decay of methanogenic archaea resulting in complete failure of the digestion process. The specific decay rate of methanogenic archaea was increased 10 times in a low pH environment (pH 5.1) comparing to neutral pH (pH 7.0) [20]. Wongfaed et al. [1] also reported that low pH and high oil content of raw POME caused the imbalance of the AD process, leading to decreased $CH_4$ yield. The final pH of 6.5–7.8 was obtained in raw POME and lower than the final pH of pH adjustment POME (7.9–8.3). Raw POME has high VFA/alkalinity ratios of 0.22 and 0.35 at initial VS loading of 13 and 20 g-VS/L, respectively. The AD process should have a VFA/alkalinity ratio of 0.10–0.30 to avoid acidification [34]. The AD process of raw POME was unstable due to the high VFA/alkalinity ratio (>0.30) [1]. POME was a suitable substrate for methane production, but the low pH of POME was not suitable for methanogenic archaea in the AD process.

**Table 3.** Biogas production from raw POME, pH adjustment POME (pH 7.5) by NaHCO$_3$, and biogas effluent.

| Substrates | Initial Loading | Methane Yield (mL-CH$_4$/gVS) | Hydrolysis Constant (d$^{-1}$) | Initial pH | Final pH | VFA/ALK Ratio |
|---|---|---|---|---|---|---|
| Raw POME (pH 4.3) | 7.5 g-VS/L | 236 | 0.038 | 6.54 | 7.8 | 0.11 |
| | 13 g-VS/L | 228 | 0.036 | 6.2 | 7.1 | 0.22 |
| | 20 g-VS/L | 125 | 0.031 | 6.1 | 6.5 | 0.35 |
| pH adjustment POME (pH 7.5) | 7.5 g-VS/L | 362 | 0.100 | 7.5 | 8.3 | 0.08 |
| | 13 g-VS/L | 345 | 0.095 | 7.5 | 8.1 | 0.11 |
| | 20 g-VS/L | 300 | 0.08 | 7.5 | 7.9 | 0.13 |
| Biogas effluent | 100% (*v/v*) | 10 | 0.08 | 8.5 | 8.8 | 0.01 |
| | 70% (*v/v*) | 12.5 | 0.05 | 8.3 | 8.7 | 0.01 |
| | 50% (*v/v*) | 18 | 0.03 | 7.5 | 8.2 | 0.01 |

The methane yields of pH adjustment POME (pH 7.5) by NaHCO$_3$ at initial VS loading of 7.5, 13, and 20 g-VS/L were 362, 345, and 300 mL-CH$_4$/gVS, respectively. The pH adjustment POME has a maximum CH$_4$ yield of 362 mL-CH$_4$/gVS, corresponding to CH$_4$ production of 21 m$^3$-CH$_4$/m$^3$-POME or approximately 35 m$^3$-biogas/m$^3$-POME. Raw POME has methane yield 2 times lower than pH adjustment POME. These results are in line with Fang et al. [35], who found that raw POME and de-oiled POME with an adjusted pH to 7 through a buffer from a BA medium has high methane yields of 503–610 mL-CH$_4$/gVS. The pH of POME was easily changed due to high oil content and the low buffering capacity resulting in an imbalance of the AD process [7] and decreasing methane production. The pH adjustment POME achieved by the addition of alkaline chemicals resulted in higher methane yields. The hydrolysis rate ($k_h$) of pH adjustment POME (0.08–0.10 d$^{-1}$) was 3 times higher than that of raw POME (0.031–0.038 d$^{-1}$). The $k_h$ value of POME (0.031–0.100 d$^{-1}$) was similar to the $k_h$ value of waste-activated sludge (0.026–0.035 d$^{-1}$) [36] under the AD process. The low pH, high COD, high oil content of POME negatively impacted on the AD process. Biogas production from biogas effluent was low due to low biodegradability organic substrate remaining. Biogas production of 100%, 70%, and 50% (*v/v*) biogas effluent was 12, 15, and 22 mL-CH$_4$/gVS, respectively. The 100% biogas effluent has no negative effect on methanogenic during the batch experiment. Biogas effluent recycling at 100% to AD showed no significant effect on the biogas production performance [37]. The 100% biogas effluent could be used as cheap alkaline material for the pH adjustment of POME before feeding to the AD reactor.

*3.2. Biogas Production from pH Adjustment POME by Oil Palm Ash and Biogas Effluent*

Methane production from POME adjusted pH by oil palm ash and biogas effluent at various concentrations are shown in Figure 2. Methane production was achieved within 14 days, indicating the easy degradability of POME. The methane yield of pH adjustment POME with oil palm ash was 3 times higher than the digestion of POME alone. Methane yields from POME adjusted pH with oil palm ash at 0, 2.5, 5, 7.5, 10, 12.5, and 15% *w/v* were 132, 375, 440, 408, 304, 293, and 235 mL-CH$_4$/gVS, respectively. The addition of oil palm ash for pH adjustment at concentrations higher than 5% *w/v* increased solids accumulation in the system, making it difficult to feed POME into the AD reactor and gain satisfactory agitation (Table 4). The addition of oil palm ash for pH adjustment at 5% (*w/v*) could supply enough buffering to maintain the AD process. The AD of pH adjustment POME with oil palm ash at 5% (*w/v*) was also satisfactory in terms of methane yield and pH, COD removal efficiency, and solids accumulation. Methane production of pH adjustment POME by 5% (*w/v*) oil palm ash was 25.5 m$^3$-CH$_4$/m$^3$-POME, corresponding to 42.5 m$^3$-biogas/ton-POME. The methane production from POME adjusted pH by 5% (*w/v*) oil palm ash (25.5 m$^3$-CH$_4$/m$^3$-POME) was significantly different ($p < 0.05$) with raw POME (13.7 m$^3$-CH$_4$/m$^3$-POME). The addition of oil palm ash could release some alkaline and trace metals, resulting in beneficial effects on the AD process. The pH adjustment POME with oil palm ash also benefited the degradation efficiency, alkalinity stability, and pH

stability. Oil palm ash had a buffer capacity that maintained the alkalinity in the AD system, leading to high biogas production. The suitable amount of oil palm ash addition to the AD reactor was 5% (*w/v*) or 1 g-ash/1 g-VS. Lo et al. [9] found that the addition of 1 g-ash/g-total solids (TS) into the AD process for the organic fraction of municipal solid waste (OFMSW) increased the biogas yield. Moreover, the addition of wood ash into the AD reactor increases the pH and prevents the unstable reactor conditions from acidification or dealing with low pH feedstock, such as rubber latex wastewater [33] or food residue [38].

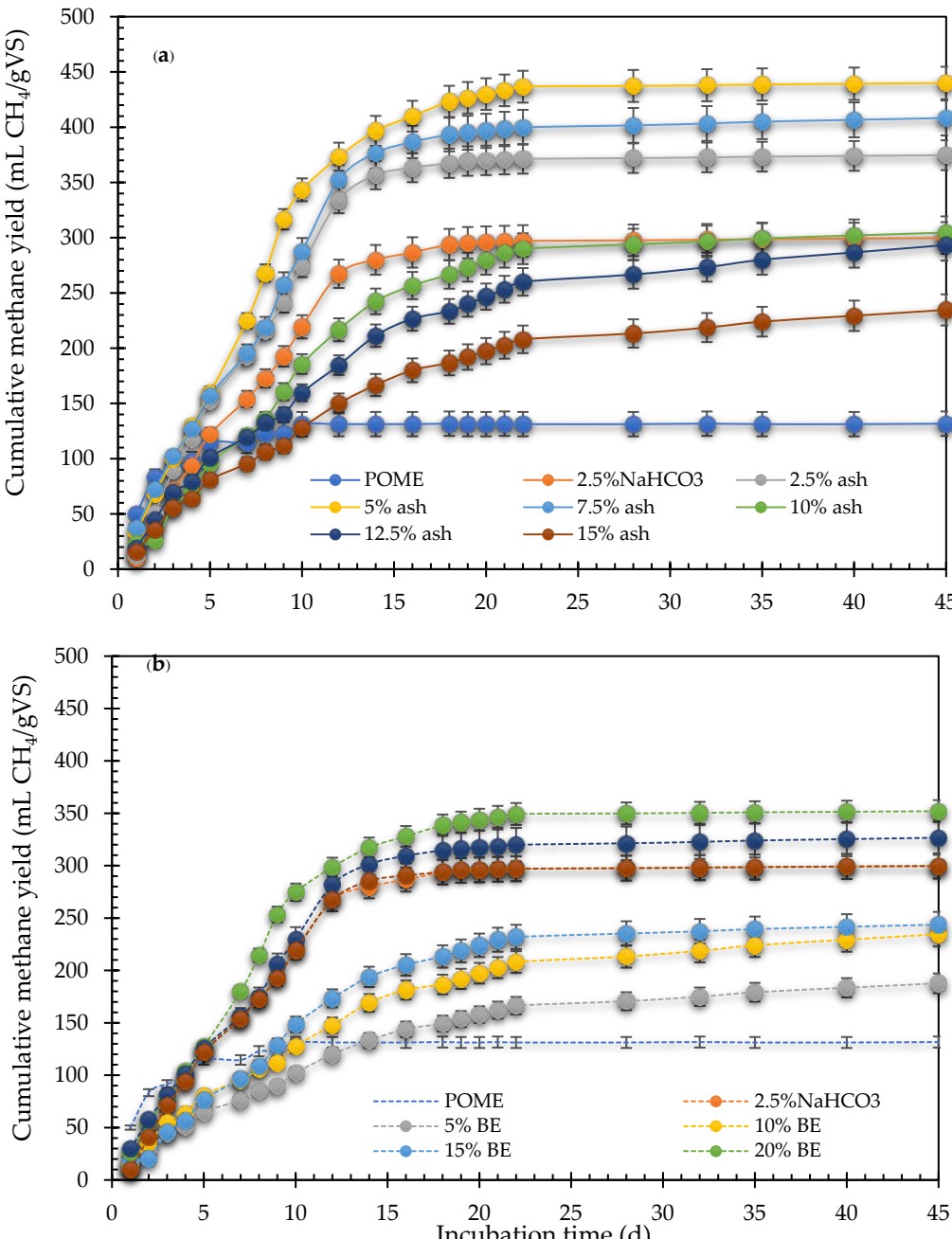

**Figure 2.** Cumulative methane yield of palm oil mill effluent adjusted pH with oil palm ash (**a**) and biogas effluent with biogas effluent (BE) (**b**) at various concentrations.

**Table 4.** Biogas production performance of pH adjustment POME by oil palm ash (ash) and biogas effluent (BE) at various concentrations.

| Conditions | Methane Yield (mL-CH$_4$/gVS) | Hydrolysis Constant (d$^{-1}$) | Solids Accumulation (g/L) | Initial pH | Final pH | COD Removal (%) |
|---|---|---|---|---|---|---|
| Raw POME | 132 | 0.031 | 37.1 | 4.8 | 6.2 | 28 |
| POME adjusted PH | | | | | | |
| 2.5% NaHCO$_3$ | 300 | 0.080 | 17.1 | 7.0 | 7.9 | 64 |
| 2.5% ash | 375 | 0.112 | 19.3 | 6.2 | 7.8 | 80 |
| 5% ash | 440 | 0.128 | 26.2 | 6.6 | 8.6 | 94 |
| 7.5% ash | 408 | 0.120 | 31.3 | 7.1 | 8.2 | 87 |
| 10% ash | 304 | 0.071 | 45.7 | 7.5 | 9.0 | 65 |
| 12.5% ash | 293 | 0.065 | 72.6 | 7.8 | 9.2 | 62 |
| 15% ash | 235 | 0.052 | 95.4 | 8.2 | 9.8 | 50 |
| 5% BE | 188 | 0.044 | 21.1 | 5.5 | 6.3 | 40 |
| 10% BE | 237 | 0.054 | 25.6 | 5.8 | 6.8 | 51 |
| 15% BE | 243 | 0.058 | 31.2 | 6.3 | 7.4 | 53 |
| 20% BE | 351 | 0.095 | 26.4 | 6.5 | 7.9 | 76 |
| 25% BE | 327 | 0.084 | 32.1 | 6.7 | 8.2 | 70 |
| 30% BE | 298 | 0.077 | 41.9 | 6.8 | 8.5 | 64 |

Methane yields from pH adjustment POME by biogas effluent at 0, 5, 10, 15, 20, 25, and 30% ($v/v$) were 132, 188, 237, 243, 351, 327, and 298 mL-CH$_4$/gVS, respectively. The addition of biogas effluent for pH adjustment has lower solid accumulation than oil palm ash. The addition of biogas effluent for pH adjustment at 20% ($v/v$) could supply enough buffering to maintain the AD process. The AD process of pH adjustment POME by biogas effluent at 20% ($v/v$) was also satisfactory in terms of methane yield and final pH, COD removal, and solids accumulation. Methane production of pH adjustment POME by 20% ($v/v$) biogas effluent was 20.4 m$^3$-CH$_4$/m$^3$-POME, corresponding to 34.0 m$^3$-biogas/ton-POME. Methane production from POME adjusted pH by 20% ($v/v$) biogas effluent (20.4 m$^3$-CH$_4$/m$^3$-POME) was significantly different ($p < 0.05$) with raw POME (13.7 m3-CH$_4$/m$^3$-POME). Biogas effluent could provide alkalinity sources for methanogenic archaea in the AD system [39]. Ladu and Lu [40] also found that biogas effluent recycling at a ratio of 2:1 increased the biogas production rate from 6.72 L/d to 7.26 L/d with more reactor stability and removal efficiencies. The biogas production from pineapple pulp and peel in a plug-flow reactor was increased by adjusting pH, increased alkalinity, and replenishment of microbes when pineapple pulp and peel were mixed biogas effluent before feeding to the AD reactor [41]. Recirculation biogas effluent can help in neutralizing the pH through dilution of incoming acidic feed stream with subsequent improvement in conversion efficiency. Recirculation of biogas effluent from the AD process can be employed to maintain a pH level favorable for methanogenic archaea resulting in high efficiency of the AD process [42]. Both oil palm ash and biogas effluent were evaluated for pH adjustment of POME before feeding into the AD reactor in a continuous process to monitor the change of microbial population in the AD reactor.

*3.3. Continuous Biogas Production from pH Adjustment POME by Oil Palm Ash Addition and Biogas Effluent Recycling*

Continuous biogas production from pH adjustment POME by 5% ($w/v$) oil palm ash (CSTR reactor R1) obtained higher stability and high methane production than pH adjustment by 20% ($v/v$) biogas effluent recycling (CSTR reactor R2). More than 90% of COD removal efficiency could be obtained from reactor R1 within two weeks. Methane production rates from pH adjustment by 5% ($w/v$) oil palm ash and 20% biogas effluent recycling were 1.27 $\pm$ 0.2 and 0.92 $\pm$ 0.12 L-CH$_4$/L/d, respectively. Methane production from POME in a continuous reactor of pH adjustment by 5% $w/v$ oil palm ash and 20% $v/v$ biogas effluent recycling was 19.1 $\pm$ 0.25 and 13.8 $\pm$ 0.3 m$^3$ CH$_4$/m3-POME, respectively, corresponding to methane yield of 328 $\pm$ 2.6 mL-CH$_4$/gVS and 237 $\pm$ 3.2 mL-CH$_4$/gVS,

respectively (Table 5). Methane production in the R1 reactor reached a maximum methane production on day 8 and remained stable until the end of the experiment. Methane production in the R2 reactor reached a maximum methane production on day 12, which gradually decreased and stabilized at the end of the experiment (Figure 3a). The methane concentration of reactor R2 decreased gradually from 65% down to 50% at the end of the experiment. In contrast, reactor R1 reached its highest methane concentration of 68% on the 14th day and stable for the whole period of operation, which was considerably higher than other studies, where only POME was used as a substrate [35]. The daily variation in the methane yield of the R1 and R2 reactor was lower than 10%, indicating the high system stability of both reactors. In contrast, reactor R1 was lower acetic acid and propionic acid accumulation than reactor R2. Reactor R1 was efficient for the conversion of VFAs to methane by methanogens. The volatile fatty acids in reactor R1 ($0.22 \pm 0.03$ g/L) were lower than reactor R2 ($0.97 \pm 0.14$ g/L). The remaining VFA in reactor effluent was acetic acid and propionic acid for both reactors. The concentration of acetic acid and propionic acid in reactor R1 were $0.08 \pm 0.01$ g/L and $0.14 \pm 0.02$ g/L, respectively (Figure 3b). The concentration of acetic acid and propionic acid in reactor R2 were $0.28 \pm 0.05$ g/L and $0.69 \pm 0.12$ g/L, respectively (Figure 3c). The pH profile of reactor R1 was more stable than reactor R2. Reactor R1 has stable pH at $7.5 \pm 0.14$ for 60 days of continuous operation. Reactor R2 has unstable pH and trend to decreased during continuous reactor operation. Reactor R2 reactor has stable pH at $6.9 \pm 0.14$ during the first 2 weeks of operation and trend to unstable after 2 weeks of operation at pH of $6.3 \pm 0.4$ (Figure 4a).

**Table 5.** Biogas production from pH adjustment POME by 5% *w/v* oil palm ash and 20% *v/v* biogas effluent recycling.

| Reactors | Methane Yield (mL-CH$_4$/gVS) | Methane Production (m$^3$ CH$_4$/m$^3$-POME) | Alkalinity (gCaCO$_3$/L) | pH | VFA (g/L) |
|---|---|---|---|---|---|
| R1—5% *w/v* oil palm ash | $328 \pm 2.6$ | $19.1 \pm 0.25$ | $2.5 \pm 0.15$ | $5.7 \pm 0.24$ | $0.22 \pm 0.03$ |
| R2—20% *v/v* biogas effluent recycling | $237 \pm 3.2$ | $13.8 \pm 0.3$ | $1.5 \pm 0.17$ | $7.8 \pm 0.23$ | $0.97 \pm 0.14$ |

The alkalinity of reactor R1 ($2.5 \pm 0.15$ g/L) was higher than reactor R2 ($1.5 \pm 0.17$ g/L). The alkalinity of reactor R1 was stable through the operation time, while the alkalinity of reactor R2 trend decreased (Figure 4b). Reactor R2 has a high volatile fatty acid/alkalinity ratio of 0.3–0.6, indicating an unstable AD process [1]. The volatile fatty acids/alkalinity ratio should be maintained in a 0.10–0.30 range to avoid acidification of the AD process [34]. The oil palm ash addition to POME for pH adjustment before feeding to the AD reactors can provide a buffer and flavor the growth of methanogenic archaea in the AD reactor. It should be noted that the oil palm ash contains metal, which is a co-enzyme in anaerobic microorganism activity [13]. The oil palm ash addition to POME for pH adjustment before feeding to the AD reactors could increase pH to flavor the growth of methanogenic archaea in the AD reactor.

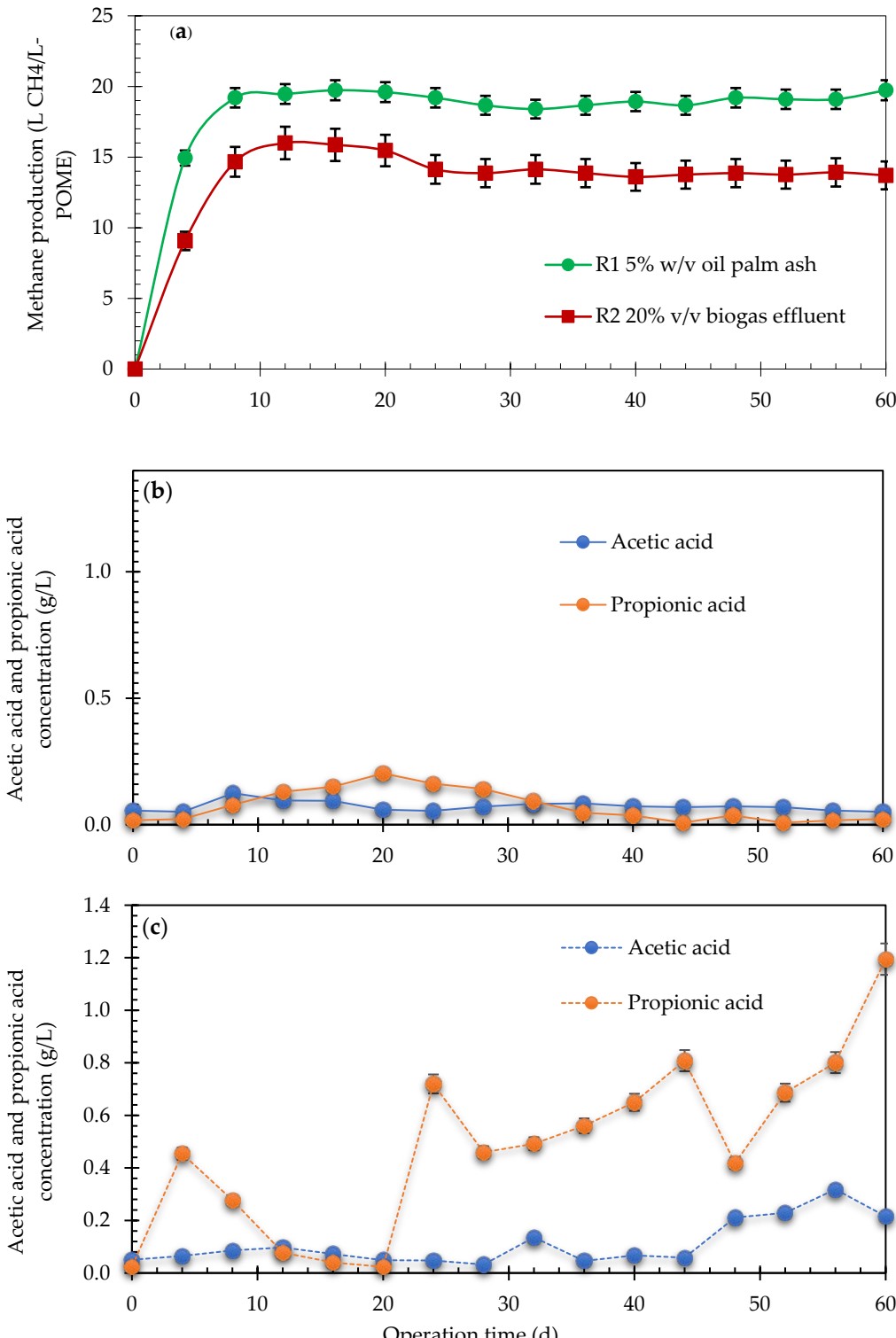

**Figure 3.** Profiles of the methane production from pH adjustment POME with oil palm ash (reactor R1) and biogas effluent recycling (reactor R2) (**a**) and change in volatile fatty acids of reactor R1 (**b**) and reactor R2 (**c**).

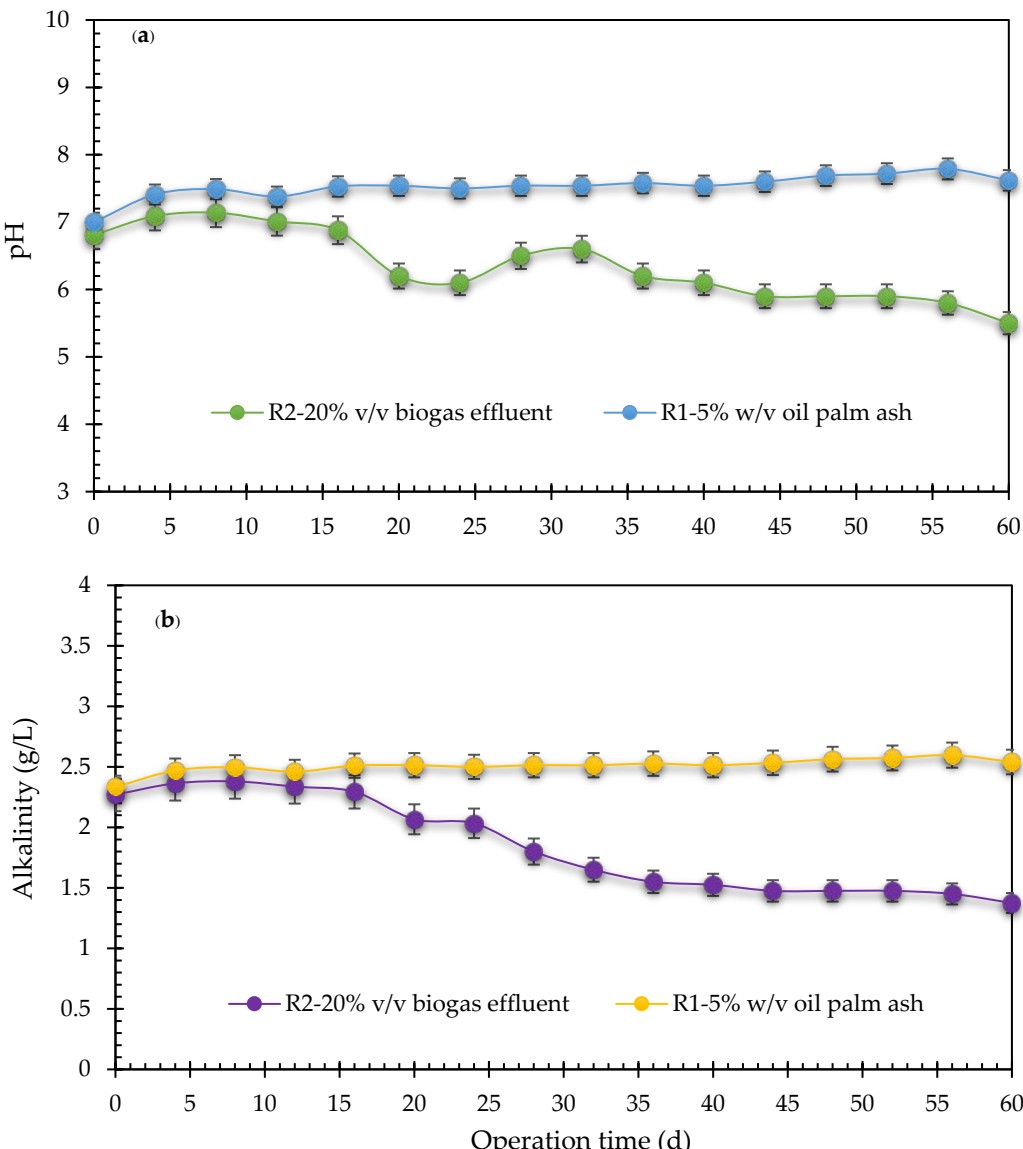

**Figure 4.** Profiles of pH (**a**) and alkalinity (**b**) during methane production from pH adjustment POME with oil palm ash and biogas effluent recycling.

### 3.4. Change of Microbial Population of pH Adjustment POME by Oil Palm Ash Addition and Biogas Effluent Recycling

The change of microbial population in reactor R1 (pH adjustment by oil palm ash) and reactor R2 (pH adjustment by biogas effluent recycling) at various operation times of 0, 7, 14, 21, 28, 35, 42 days were monitoring for effect on number bacteria and archaea. The bacteria population was similar for reactor R1 and R2 (Figure 5a). The number of bacteria had a slight change even during the low pH environment. Sun et al. [20] found that the reduction in the number of bacteria was only 25% during low pH inhibition. Reactor R1 and R2 were dominated by *Tepidanaerobacter* sp., *Thermoanaerobacterium* sp., *Anaerobaculum* sp., *Clostridium* sp., and *Petrotoga* sp. *Petrotoga* sp. can produce formic acid, acetic acid, lactic acid, butyric acid, and caproic acid from various carbohydrates. *Anaerobaculum* sp. is responsible for protein fermentation, while *Tepidanaerobacter* sp. can degrade alcohol and lactic acid [43]. *Thermoanaerobacterium* sp. is commonly found in thermophilic anaerobic digestion systems. *Thermoanaerobacterium* sp. could convert carbohydrates to hydrogen with acetic and butyric acid as the end product [44]. The bacteria communities

in reactor R1 consisted of *Pseudomonas* sp., *Petrotoga* sp., *Clostridium* sp., Uncultured *Clostidiales*, *Thermoanaerobacterium* sp., *Actinobacterium* sp., *Tepidimicrobium* sp., *Clostridium* sp., *Anaerobaculum* sp., and *Tepidanaerobacter* sp. The bacteria communities in the R2 reactor consisted of *Pseudomonas* sp., *Petrotoga* sp., *Thermoanaerobacterium* sp., *Actinobacterium* sp., *Tepidimicrobium* sp., *Anaerobaculum* sp., and *Tepidanaerobacter* sp. (Figure 5a).

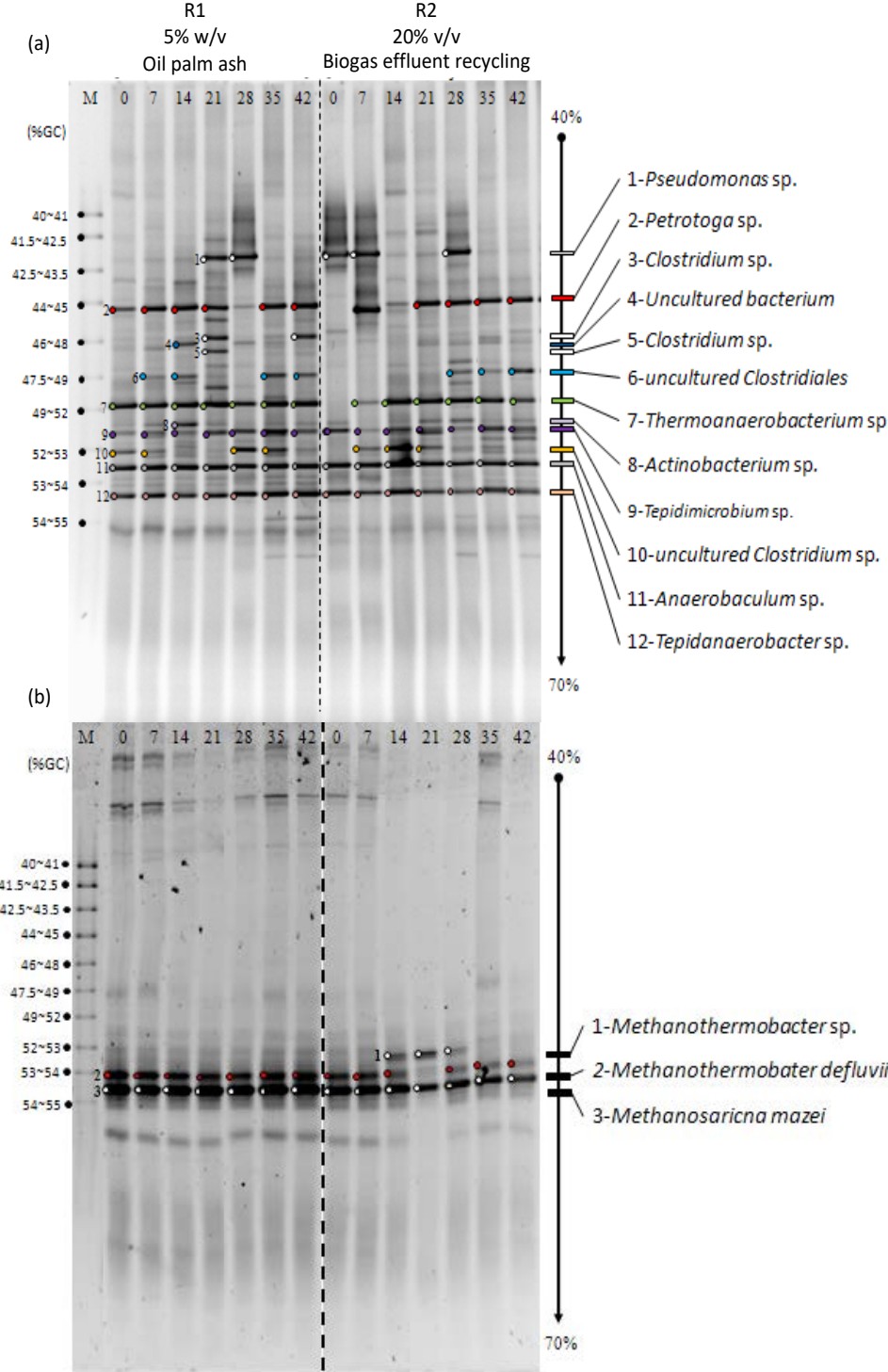

**Figure 5.** Microbial dynamic population bacteria (**a**) and archaea (**b**) in the AD of from pH adjustment POME by oil palm ash (R1) and biogas effluent recycling (R2) analyzed by PCR-DGGE. M was a DGGE marker and numbers referred to days of reactor operation.

The reactor R1 had a higher methane production and reactor stability than reactor R2, corresponding to the high stability of the archaea community in reactor R1. The archaea diversity was similar in both reactors, while the number of archaea in reactor R1 was higher than reactor R2, indicating a strong DGGE band (Figure 5b). The dynamic population of the archaea in the R2 reactor was unstable after 2 weeks of operation, corresponding with the unstable pH and alkalinity performance. Even reactor R2 has stability in biogas production, but the unstable pH and alkalinity leading the reactor R2 have a high potential to inhibit by low pH or pH drop. Meanwhile, the dynamic population of the bacterial community did not significantly differ in either reactor. The results indicated that the pH adjustment POME by 5% (*w/v*) oil palm ash had a more substantial influence on the archaeal community than the bacterial community. The pH adjustment by oil palm ash addition enhanced the number of *Methanothermobacter defluvii* and *Methanosarcina mazei* corresponding to high methane production. The methanogenic archaea community in reactor R1 comprised *Methanothermobacter defluvii* and *Methanosarcina mazei*. Meanwhile, the methanogenic archaea in reactor R2 comprised only *Methanosarcina mazei* (Figure 5b). *Methanothermobacter defluvii* can reduce $CO_2$ with hydrogen, formic acid, or acetic acid as terminal electron acceptors to methane [44]. *Methanosarcina* sp. are organisms that grow at a temperature range between 50–60 °C. Genus *Methanosarcina* can tolerate high acetic acid concentrations and utilize acetic acid as a substrate [45,46]. A *Methanosarcina* population can enable system stability and high methane production. Podmirseg et al. [47] also found that the addition of wood ash into anaerobic reactors had a more significant influence on archaeal activity than bacterial activity, while Methanosarcina was the most dominant methanogen in a reactor with wood ash addition.

The pH adjustment of POME by oil palm ash enhanced biogas production by an increased buffer capacity and increased the number of methanogenic archaea. The pH adjustment by oil palm ash addition had a higher methane production for the long-term operation with the stability of pH, gas production, alkalinity, and archaea community. Oil palm ash is a cost-effective alkali material for enhancing AD reactor stability and abundantly available at palm oil mill plants.

## 4. Conclusions

The low pH, high easily biodegradable material, high oil content, and low alkalinity of POME were easily created a low pH environment in the AD reactor resulting in complete failure of the biogas production process. Oil palm ash is a cost-effective alkali material for pH adjustment of POME as a buffer and trace metals source, resulting in preventing the pH drop and the increased methanogen population in the AD process. The optimal dose for pH adjustment by oil palm ash addition was 5% *w/v* with a methane yield of 440 mL-$CH_4$/gVS. The pH adjustment of POME by 5% *w/v* oil palm ash can prevent the drop of pH in a continuous AD process by enhancing the stability of pH, gas production, alkalinity, archaea community, and biogas production. Methane production and methane yield from POME in a continuous reactor of pH adjustment by 5% *w/v* oil palm ash was $19.1 \pm 0.25$ m$^3$ $CH_4$/m$^3$-POME and $328 \pm 2.6$ mL-$CH_4$/gVS, respectively. The pH adjustment of POME by oil palm ash was enhanced biogas production by an increased buffer capacity and increased the number of *Methanothermobacter defluvii* and *Methanosarcina mazei*.

**Author Contributions:** Conceptualization, A.S., C.M. and S.O.-T.; methodology, A.S.; validation, A.S., C.M. and S.O.-T.; formal analysis, A.S. and C.M.; investigation, A.S., C.M., A.R. and S.O.-T.; resources, A.R. and S.O.-T.; data curation, A.S., C.M. and S.O.-T.; writing—original draft preparation, A.S., C.M. and S.O.-T.; writing—review and editing, A.S., C.M. and S.O.-T.; visualization, A.S. and C.M.; supervision, A.R. and S.O.-T.; funding acquisition, S.O.-T. and A.R. All authors have read and agreed to the published version of the manuscript.

**Funding:** This research was funded by The Royal Golden Jubilee Ph.D. Program (RGJ-Ph.D. Program), grant number PHD57/0042, The Energy Policy and Planning Office (EPPO), grant number 044/2560, Thailand Research Fund for Senior Research Scholar, grant number RTA6280001 and Mid-Career Research Grant, grant number RSA6180048.

**Institutional Review Board Statement:** Not applicable.

**Informed Consent Statement:** Not applicable.

**Data Availability Statement:** Not applicable.

**Acknowledgments:** All the authors of the manuscript are immensely grateful to the Research and Development Institute Thaksin University for their technical assistance and valuable support in completing this research project.

**Conflicts of Interest:** The authors declare that there is no conflict of interest regarding the publication of this paper.

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
