# Peer review of "Enhancement of Thermophilic Biogas Production from Palm Oil Mill Effluent by pH Adjustment and Effluent Recycling"

_processes, doi:10.3390/pr9050878_

Round 1
Reviewer 1 Report
This work explores the possibility of using palm oil ash to adjust the pH of a palm oil mill effluent (POME) during biogas production. Besides, the recycling of part of the biogas produced was also considered to adjust the pH. This investigation is very timely and interesting, the manuscript is nicely written, and the experimental data support the conclusions. Therefore, I strongly support the publication of this work as it is.
Author Response
Response to reviewers
Manuscript ID: processes-1196430
Type of manuscript: Article
Title: Enhancement of thermophilic biogas production from palm oil mill
effluent by pH adjustment and effluent recycling
Reviewer #1: This work explores the possibility of using palm oil ash to adjust the pH of a palm oil mill effluent (POME) during biogas production. Besides, the recycling of part of the biogas produced was also considered to adjust the pH. This investigation is very timely and interesting, the manuscript is nicely written, and the experimental data support the conclusions. Therefore, I strongly support the publication of this work as it is.
Response: Thank you very much for the positive review.

Reviewer 2 Report
Manuscript review: Enhancement of thermophilic biogas production from palm oil mill effluent by pH adjustment and effluent recycling (processes-1196430).
The manuscript deals with the impact of different doses of palm oil ash and gaseous wastewater recycling on the prevention of low pH and rapid acidification.
- The abstract should be redrafted. It is incomprehensible. It will be more understandable if it includes: the sentence of introduction with justification of the presented studies, the purpose of the work, the test methods and the results obtained.
- A review of literature is absolutely not enough. Please highlight the knowledge gap, justify the purpose of your research, and write how it differs from other authors? The literature presented in the manuscript does not include the latest publication achievements. Most of the literature is from ten years ago.
- Line 84. The test material is not sufficiently described. Please describe more the parameters of the raw materials from which the palm mill waste water (POME) was obtained.Linia 143.
- The mathematical model is missing. One equation is not a mathematical model.
- Please indicate the parameters of the test apparatus used.
- The conclusions of the studies carried out need to be supplemented.
- There is a lack of novelty elements in relation to a good review of literature. Especially since in recent years a lot of manuscripts on this subject have been published.
- The experiment should be expanded, computer simulations added. This would allow other parameters of the experiment to be considered.
Author Response
Reviewer #2: Manuscript review: Enhancement of thermophilic biogas production from palm oil mill effluent by pH adjustment and effluent recycling (processes-1196430). The manuscript deals with the impact of different doses of palm oil ash and gaseous wastewater recycling on the prevention of low pH and rapid acidification.
- The abstract should be redrafted. It is incomprehensible. It will be more understandable if it includes: the sentence of introduction with justification of the presented studies, the purpose of the work, the test methods and the results obtained.
Response: The abstract was rewritten as a reviewer suggestion.
“The sudden pH drops always inhibit the AD reactor for biogas production from palm oil mill effluent (POME). The pH adjustment of POME by oil palm ash addition and biogas effluent recycling was investigated for preventing the drop of pH. The pH adjustment POME to 7.5 was increased methane yield 2 times than raw POME (pH 4.3). The pH adjustment by oil palm ash addition at 5% w/v was optimal dose with methane yield of 440 mL-CH4/gVS. The pH adjustment by biogas effluent recycling at 20% v/v was optimal dose with methane yield of 351 mL-CH4/gVS. Methane production from POME in a continuous reactor of pH adjustment by 5% w/v oil palm ash and 20% v/v biogas effluent recycling was 19.1±0.25 and 13.8±0.3 m3 CH4/m3-POME, respectively, corresponding to methane yield of 328±2.6 mL-CH4/gVS and 237±3.2 mL-CH4/gVS, respectively. The pH adjustment by oil palm ash has higher methane production than the biogas effluent recycling for the long-term operation with the stability of pH, alkalinity, and archaea community. The pH adjustment by oil palm ash increased the number of Methanosarcina mazei and Methanothermobacter defluvii. Oil palm ash is a cost-effective alkali material for preventing pH drop and enhancing AD reactor stability.”
- A review of the literature is absolutely not enough. Please highlight the knowledge gap, justify the purpose of your research, and write how it differs from other authors? The literature presented in the manuscript does not include the latest publication achievements. Most of the literature is from ten years ago.
Response: The updated literature and highlight the knowledge gap was added
Palm oil mill effluent (POME) is the main wastewater generated from the palm oil extraction plant, which is low pH (4.9), high chemical oxygen demand (68 g/L), and high oil content (6.2 g/L)[1]. The 0.5 m3 of POME was generated from the processing of 1 tonne of palm oil fresh fruit bunches [2]. It is estimated that 400–600 m3/d was generated from the mill plant with a processing capacity of 800-1000 tonne/d.
The oil palm biomass is often used as boiler fuel by palm oil mill plants to produce steam for electricity generation and palm oil extraction, generating many oil palm ash [12]. The addition of oil palm ash at 8.0% w/v can recover the inhibited sludge from a full-scale biogas reactor with increased pH and buffer capacity under mesophilic conditions [13]. The addition of ash can act as a pH adjustment in the AD reactor by enhancing the microbial growth rate with the release of alkali and trace metals [14].
However, the addition of OPA and biogas effluent recycling for pH adjustment of POME to improve the stability of AD reactor and biogas production under thermophilic conditions has never been studied. Proper amounts of oil palm ash addition and biogas effluent recycling into the AD reactors without disturbing the microbial communities or biochemical processes are required.
This work investigated the pH adjustment of POME by oil palm ash addition and effluent recycling as a pH regulator to improve the stability and feasibility of the process. The effects of pH adjustment of POME by oil palm ash addition and effluent recycling on change in dynamic bacteria and archaea population were investigated.
- Line 84. The test material is not sufficiently described. Please describe more the parameters of the raw materials from which the palm mill wastewater (POME) was obtained. Lin 143.
Response: The explanation of test materials was added.
“POME and biogas effluent was measured for alkalinity, pH, total volatile fatty acids, COD, total nitrogen (TN), total solids (TS), volatile solids (VS), and oil and grease. Oil palm ash was measured for total solids (TS), pH, and alkalinity.”
- The mathematical model is missing. One equation is not a mathematical model.
Response: The equation used to explain the only kinetic value to determine the biodegradation rate.
- Please indicate the parameters of the test apparatus used.
Response: The test parameter for the continuous reactor was added
“Biogas volume and pH were recorded daily, while volatile fatty acid concentrations were measured every four days. The H2, CO2, and CH4 in biogas were analyzed daily by gas chromatography. The effluent samples from the reactors were analyzed for VFA, pH, and alkalinity. The sludge samples were taken every week from both reactors for dynamic microbial population analysis.”
- The conclusions of the studies carried out need to be supplemented.
Response: The conclusion was corrected
“Oil palm ash is a cheap alkali material suitable for the pH adjustment of POME before feeding to the AD process with high reactor stability and high methane production. Methane production from POME in a continuous reactor of pH adjustment by 5% w/v oil palm ash and 20% v/v biogas effluent recycling was 19.1±0.25 and 13.8±0.3 m3 CH4/m3-POME, respectively, corresponding to methane yield of 328±2.6 mL-CH4/gVS and 237±3.2 mL-CH4/gVS, respectively. The pH adjustment by oil palm ash addition enhanced the number of Methanothermobacter defluvii and Methanosarcina mazei corresponding to high methane production. The pH adjustment of POME by 5% w/v oil palm ash can prevent the drop of pH during batch and continuous AD process with enhancing the stability of pH, gas production, alkalinity, archaea community, and biogas production. The pH adjustment of POME by 20% v/v biogas effluent recycling can prevent the drop of pH during the batch and continuous AD process with enhancing gas production but unstable in terms of pH, alkalinity, archaea community, and volatile fatty acids concentration. The pH adjustment by oil palm ash addition has higher methane production than the biogas effluent recycling for the long-term operation with the stability of pH, gas production, alkalinity, and archaea community. Oil palm ash is a cost-effective alkali material for enhancing AD reactor stability and abundantly available at palm oil mill plants.”
- There is a lack of novelty elements in relation to a good review of the literature. Especially since in recent years, a lot of manuscripts on this subject have been published.
Response: The gap of knowledge was added
“The POME had high organic content in terms of COD and high temperature (80-92°C). Thermophilic AD is suitable for POME due to low cooling cost requirements, and also increases in the rates of chemical and enzymatic reactions, an increase in thermodynamic favorability of methane production reactions [21]. However, the addition of OPA and biogas effluent recycling for pH adjustment of POME to improve the stability of AD reactor and biogas production under thermophilic conditions has never been studied. Proper amounts of oil palm ash addition and biogas effluent recycling into the AD reactors without disturbing the microbial communities or biochemical processes are required.
This work investigated the pH adjustment of POME by oil palm ash addition and effluent recycling as a pH regulator to improve the stability and feasibility of the process. The effects of pH adjustment of POME by oil palm ash addition and effluent recycling on change in dynamic bacteria and archaea population were investigated.”
- The experiment should be expanded, computer simulations added. This would allow other parameters of the experiment to be considered.
Response: I agree with the reviewer that computer simulations should give valuable information. However, computer simulations it not our expertise. The authors try to link the performance with the microorganism to allow the parameter for mean in terms of performance and response microorganisms inside the AD reactor.

Reviewer 3 Report
The manuscript presents a study on the enhancement of thermophilic biogas production from palm oil mill effluent (POME) by palm oil biomass ash and biogas effluent recycling.
Corrections to minor methodological errors and text editing are required.
More specifically, the following issues should be considered:
- Lines 15, 51, 104 subscripts are missing.
- Line 34, 41, 60, 167 acronym AD was not used.
Materials and methods
- Lines 110-111: please, specify how the Authors expressed the substrate to inoculum ratio (gVS/gVS?)
- Paragraph 2.2: please, specify the working volume of the batch reactor.
- Please, specify if the volume of biogas /methane was converted to standard temperature and pressure conditions (T = 273.15 K, P = 105 Pa).
Results
- Please, define the volatile solids used to express the specific methane production (VSmixture or VSsubstrate)
- Line 189: please, replace ‘imbalanced’ with ‘instable’.
- Table 1: how the VFAs were expressed (e. g.: gC/l or gAcetate/l)
- Table 3: the chemical constituents of palm oil biomass were expressed in term of %, but it is not clear how the percentage was expressed ( % of dry weight of sample?). In addition, why the sum of all the constituents is not equal to 100?
- What about the methane production yield in continuous reactor operation?
Author Response
Reviewer #3: The manuscript presents a study on the enhancement of thermophilic biogas production from palm oil mill effluent (POME) by palm oil biomass ash and biogas effluent recycling.
Corrections to minor methodological errors and text editing are required.
More specifically, the following issues should be considered:
- Lines 15, 51, 104 subscripts are missing.
Response: The whole manuscript was edited and revised as suggested.
- Line 34, 41, 60, 167 acronym AD was not used.
Response: The anaerobic digestion (AD) was replaced by the acronym AD.
Materials and methods
- Lines 110-111: please, specify how the Authors expressed the substrate to inoculum ratio (gVS/gVS?)
Response: The substrate to inoculum (S/I) was 2:1 based on VS basis of POME and anaerobically digested sludge. (please see Line 113-114).
- Paragraph 2.2: please, specify the working volume of the batch reactor.
Response: The experiments were performed in 500 mL glass serum bottles with a working volume of 300 mL was added (please see Line 105-106).
- Please, specify if the volume of biogas /methane was converted to standard temperature and pressure conditions (T = 273.15 K, P = 105 Pa).
Response: The following equation (Eq. 1) has been used to convert methane volume at 55 °C to standard temperature and pressure (STP) conditions (T = 273.15 K, P = 105 Pa).
(Eq. 1)
Where V1, V2, T1, and T2 were methane volume at 55 ℃ (L), methane volume at STP (L), incubation temperature (273.15+55 K), and temperature of STP (273.15 K), respectively (Tukanghan et al., 2021).
Results
- Please, define the volatile solids used to express the specific methane production (VS mixture or VS substrate)
Response: The volatile solids of the substrate was used to express the methane yield.
“The methane yield was calculated from the volatile solids of the substrate. The methane production from biogas effluent alone was subtracted from the methane production from pH adjustment by biogas effluent experiment for methane yield calculation.”
- Line 189: please, replace ‘imbalanced’ with ‘instable’.
Response: The imbalance was replaced by unstable.
- Table 1: how the VFAs were expressed (e. g.: gC/l or gAcetate/l)
Response: Volatile fatty acids concentration were expressed as g-acetic acid/L and g-propionic acid/L. The X-axis was corrected to acetic acid and propionic acid concentration (g/L)
- Table 3: the chemical constituents of palm oil biomass were expressed in terms of %, but it is not clear how the percentage was expressed (% of the dry weight of sample?). In addition, why the sum of all the constituents is not equal to 100?
Response: The unit expressed as a term of % dry weight. The unit was corrected to % dry weight.
- What about the methane production yield in continuous reactor operation?
Response: The methane yield at the steady-state of the continuous reactor was added in Table 5.

Reviewer 4 Report
This MS provides some experimental investigation on the use of oil palm ash and biogas effluent to stabilize pH of POME for biogas production. The series of experiments gave clear results, which can improve our understanding on the field. However, some revision is necessary to improve this MS before potential publication.
- Please introduce where and how much the oil palm ash is generated in the Introduction. Also, there are some instances where a different term is used (such as palm oil biomass ash in the Abstract) – please be consistent, or be specific if the terms indicate different materials.
- The long, repeated expressions such as “POME adjusted pH with oil palm ash” and “POME adjusted pH with biogas effluent” limit the readability.
- L51-52. 400-600 m3/d from one specific plant? Please give some more details.
- “POME adjusted pH with oil palm ash” or “biogas effluent” – what were the actual pH regarding different doses?
- L132. v/v?
- L152-154. Please give some more details of the PCR-DGGE and the following sequencing methods.
- Table 1. TS unit of g/kg also for POME and biogas effluent? Or g/L?
- Table 2. How was the initial pH neutralized for raw POME runs?
- Table 4. Does the methane yield for BE runs consider VS derived from biogas effluents?
- Figure 5. Why do the two initial data (0 d of ash and BE) differ for bacteria?
Author Response
Reviewer #4: This MS provides some experimental investigation on using oil palm ash and biogas effluent to stabilize the pH of POME for biogas production. The series of experiments gave clear results, which can improve our understanding on the field. However, some revision is necessary to improve this MS before potential publication.
- Please introduce where and how much the oil palm ash is generated in the Introduction. Also, there are some instances where a different term is used (such as palm oil biomass ash in the Abstract) – please be consistent, or be specific if the terms indicate different materials.
Response: The explanation of oil palm ash was added. The term of oil palm ash was used throughout the whole manuscript.
“The oil palm biomass often used as boiler fuel by palm oil mill plants to produce steam for electricity generation and palm oil extraction which is generated large amount oil palm ash [Yin et al., 2008].”
- The long, repeated expressions such as “POME adjusted pH with oil palm ash” and “POME adjusted pH with biogas effluent” limit the readability.
Response: POME adjusted pH with oil palm ash” and “POME adjusted pH with biogas effluent” were shortened to pH adjustment and effluent recycling.
- L51-52. 400-600 m3/d from one specific plant? Please give some more details.
Response: The sentence was corrected
“The 0.5 m3 of POME was generated from the processing 1 of fresh oil palm fruit [Ahmed et al., 2015]. It is estimated that 400–600 m3/d was generated from mills plant capacity of 800-1000 tonne/d.”
- “POME adjusted pH with oil palm ash” or “biogas effluent” – what was the actual pH regarding different doses?
Response: The initial pH of the different dose was added in table 4, The 5% w/v oil palm ash and 20% v/v biogas effluent has an initial pH of 6.6 and 6.5, respectively.
- L132. v/v?
Response: was corrected to % v/v
- L152-154. Please give some more details of the PCR-DGGE and the following sequencing methods.
Response: 2.5. Microbial community analysis
“Polymerase chain reaction-denaturing gradient gel electrophoresis (PCR-DGGE) was used to study microbial community structure in the reactor R1 and R2. Sludge samples were collected from bioreactors at days 0, 7, 14, 21, 28, 35, and 42 of operation times. Genomic DNA extractions and PCR-DGGE for bacteria were made as previously described [32]. DGGE profiles were compared using the Quantity One software package (version 4.6.0; Bio-Rad Laboratories). Most of the bands were excised from the gel and re-amplified. After re-amplification, PCR products were purified and sequenced by Macrogen Inc. (Seoul, Korea). The closest matches for partial 16S rRNA gene sequences were identified by database searches in Gene Bank using BLAST were added (please see Line 171-180).”
- Table 1. TS unit of g/kg also for POME and biogas effluent? Or g/L?
Response: The TS unit of POME and biogas effluent was corrected to g/L, The TS unit of oil palm ash was g/kg. The unit of TS was changed as table 1.
- Table 2. How was the initial pH neutralized for raw POME runs?
Response: The initial pH value of raw POME runs was 6.2 after mixed with inoculum. The buffer in inoculum neutralized for raw POME runs.
- Table 4. Does the methane yield for BE runs consider VS derived from biogas effluents?
Response: the methane yield for BE runs does not consider the VS from biogas effluent. The methane yield for BE runs was subtracted by BE alone from table 2. The sentence below was added in methods.
“The methane yield was calculated from the volatile solids of the substrate. The methane production from biogas effluent alone was subtracted from the methane production from pH adjustment by biogas effluent experiment for methane yield calculation.”
- Figure 5. Why do the two initial data (0 d of ash and BE) differ for bacteria?
Response: It the same inoculum but small differences in microorganisms due to the BE contain some microorganisms from biogas effluent.

Reviewer 5 Report
The manuscript addresses an interesting topic about thermophilic biogas production from palm oil mill effluent. The experimental design, the theoretical methods and the execution of the study are adequate. The discussion of the results is clear.
The paper has a good scientific soundness and deserves to be published after some modifications that are required and listed in the following text.
- The use of English is quite good; however, I suggest a fast revision of the manuscript, in order to correct some mistakes of syntax
- Keywords should be in alphabetical order.
I recommend this manuscript for publication in "Processes" Journal after completing some minor improvements
Author Response
Reviewer #5: The manuscript addresses an interesting topic about thermophilic biogas production from palm oil mill effluent. The experimental design, the theoretical methods and the execution of the study are adequate. The discussion of the results is clear.
The paper has a good scientific soundness and deserves to be published after some modifications that are required and listed in the following text.
- The use of English is quite good; however, I suggest a fast revision of the manuscript, in order to correct some mistakes of syntax
Response: the language error was checked through the whole manuscript.
- Keywords should be in alphabetical order.
Response: Thank you very much for your suggestion. The keywords were changed accordingly.
I recommend this manuscript for publication in "Processes" Journal after completing some minor improvements
Response: Thank you very much for the positive response.

Round 2
Reviewer 2 Report
Second review: Manuscript review: Enhancement of thermophilic biogas production from palm oil mill effluent by pH adjustment and effluent recycling (processes-1196430).
The manuscript deals with the impact of different doses of palm oil ash and gaseous wastewater recycling on the prevention of low pH and rapid acidification.
- Abstract corrected according to my instructions.
(WAS: The abstract should be redrafted. It is incomprehensible. It will be more understandable if it includes: the sentence of introduction with justification of the presented studies, the purpose of the work, the test methods and the results obtained.)
- The literature review still needs improvement. No significant improvements have been made as I suggested.
(WAS: A review of literature is absolutely not enough. Please highlight the knowledge gap, justify the purpose of your research, and write how it differs from other authors? The literature presented in the manuscript does not include the latest publication achievements. Most of the literature is from ten years ago.)
- An answer was given not to my question. I asked about the parameters of the raw materials.
(WAS: Line 84. The test material is not sufficiently described. Please describe more the parameters of the raw materials from which the palm mill waste water (POME) was obtained.)
- In that case, the mathematical model is missing. Please at least show yourself in the literature where the mathematical models you rely on are.
(WAS: The mathematical model is missing. One equation is not a mathematical model.)
- That doesn't answer my question.
(WAS: Please indicate the parameters of the test apparatus used.)
- The proposals have been improved, but still lack novelty effects in the state of knowledge.
(WAS: The conclusions of the studies carried out need to be supplemented.)
- My question has been partially answered. What does dynamic investigation mean?
(WAS: There is a lack of novelty elements in relation to a good review of literature. Especially since in recent years a lot of manuscripts on this subject have been published.)
The experiment should be expanded, computer simulations added. This would allow other parameters of the experiment to be considered.
Author Response
Reviewer #2: Manuscript review: Enhancement of thermophilic biogas production from palm oil mill effluent by pH adjustment and effluent recycling (processes-1196430). The manuscript deals with the impact of different doses of palm oil ash and gaseous wastewater recycling on the prevention of low pH and rapid acidification.
- The abstract should be redrafted. It is incomprehensible. It will be more understandable if it includes: the sentence of introduction with justification of the presented studies, the purpose of the work, the test methods and the results obtained.
Response: The abstract was rewritten as a reviewer suggestion. Page 1 line 13-25
“The sudden pH drops always inhibit the AD reactor for biogas production from palm oil mill effluent (POME). The pH adjustment of POME by oil palm ash addition and biogas effluent recycling was investigated for preventing the drop of pH. The pH adjustment POME to 7.5 was increased methane yield 2 times than raw POME (pH 4.3). The pH adjustment by oil palm ash addition at 5% w/v was optimal dose with methane yield of 440 mL-CH4/gVS. The pH adjustment by biogas effluent recycling at 20% v/v was optimal dose with methane yield of 351 mL-CH4/gVS. Methane production from POME in a continuous reactor of pH adjustment by 5% w/v oil palm ash and 20% v/v biogas effluent recycling was 19.1±0.25 and 13.8±0.3 m3 CH4/m3-POME, respectively, corresponding to methane yield of 328±2.6 mL-CH4/gVS and 237±3.2 mL-CH4/gVS, respectively. The pH adjustment by oil palm ash has higher methane production than the biogas effluent recycling for the long-term operation with the stability of pH, alkalinity, and archaea community. The pH adjustment by oil palm ash increased the number of Methanosarcina mazei and Methanothermobacter defluvii. Oil palm ash is a cost-effective alkali material for preventing pH drop and enhancing AD reactor stability.”
- A review of the literature is absolutely not enough. Please highlight the knowledge gap, justify the purpose of your research, and write how it differs from other authors? The literature presented in the manuscript does not include the latest publication achievements. Most of the literature is from ten years ago.
Response: The updated literature and highlight the knowledge gap was added Page 1 line 30-33
Palm oil mill effluent (POME) is the main wastewater generated from the palm oil extraction plant, which is low pH (4.9), high chemical oxygen demand (68 g/L), and high oil content (6.2 g/L)[1]. The 0.5 m3 of POME was generated from the processing of 1 tonne of palm oil fresh fruit bunches [2]. It is estimated that 400–600 m3/d was generated from the mill plant with a processing capacity of 800-1000 tonne/d.
Page 2 line 56-62
The oil palm biomass is often used as boiler fuel by palm oil mill plants to produce steam for electricity generation and palm oil extraction, generating many oil palm ash [12]. The addition of oil palm ash at 8.0% w/v can recover the inhibited sludge from a full-scale biogas reactor with increased pH and buffer capacity under mesophilic conditions [13]. The addition of ash can act as a pH adjustment in the AD reactor by enhancing the microbial growth rate with the release of alkali and trace metals [14].
Page 2 line 74-82
However, the addition of OPA and biogas effluent recycling for pH adjustment of POME to improve the stability of AD reactor and biogas production under thermophilic conditions has never been studied. Proper amounts of oil palm ash addition and biogas effluent recycling into the AD reactors without disturbing the microbial communities or biochemical processes are required.
This work investigated the pH adjustment of POME by oil palm ash addition and effluent recycling as a pH regulator to improve the stability and feasibility of the process. The effects of pH adjustment of POME by oil palm ash addition and effluent recycling on change in dynamic bacteria and archaea population were investigated.
- Line 84. The test material is not sufficiently described. Please describe more the parameters of the raw materials from which the palm mill wastewater (POME) was obtained. Lin 143.
Response: The explanation of test materials was added. Page 3 line 97-100
“POME and biogas effluent was measured for alkalinity, pH, total volatile fatty acids, COD, total nitrogen (TN), total solids (TS), volatile solids (VS), and oil and grease. Oil palm ash was measured for total solids (TS), pH, and alkalinity.”
- The mathematical model is missing. One equation is not a mathematical model.
Response: The equation used to explain the only kinetic value to determine the biodegradation rate. Page 4 line 157-158
- Please indicate the parameters of the test apparatus used.
Response: The test parameter for the continuous reactor was added. Page 4 line 137-142
“Biogas volume and pH were recorded daily, while volatile fatty acid concentrations were measured every four days. The H2, CO2, and CH4 in biogas were analyzed daily by gas chromatography. The effluent samples from the reactors were analyzed for VFA, pH, and alkalinity. The sludge samples were taken every week from both reactors for dynamic microbial population analysis.”
- The conclusions of the studies carried out need to be supplemented.
Response: The conclusion was corrected. Page 14 line 357-373
“Oil palm ash is a cheap alkali material suitable for the pH adjustment of POME before feeding to the AD process with high reactor stability and high methane production. Methane production from POME in a continuous reactor of pH adjustment by 5% w/v oil palm ash and 20% v/v biogas effluent recycling was 19.1±0.25 and 13.8±0.3 m3 CH4/m3-POME, respectively, corresponding to methane yield of 328±2.6 mL-CH4/gVS and 237±3.2 mL-CH4/gVS, respectively. The pH adjustment by oil palm ash addition enhanced the number of Methanothermobacter defluvii and Methanosarcina mazei corresponding to high methane production. The pH adjustment of POME by 5% w/v oil palm ash can prevent the drop of pH during batch and continuous AD process with enhancing the stability of pH, gas production, alkalinity, archaea community, and biogas production. The pH adjustment of POME by 20% v/v biogas effluent recycling can prevent the drop of pH during the batch and continuous AD process with enhancing gas production but unstable in terms of pH, alkalinity, archaea community, and volatile fatty acids concentration. The pH adjustment by oil palm ash addition has higher methane production than the biogas effluent recycling for the long-term operation with the stability of pH, gas production, alkalinity, and archaea community. Oil palm ash is a cost-effective alkali material for enhancing AD reactor stability and abundantly available at palm oil mill plants.”
- There is a lack of novelty elements in relation to a good review of the literature. Especially since in recent years, a lot of manuscripts on this subject have been published.
Response: The gap of knowledge was added. Page 2 line 74-82
“The POME had high organic content in terms of COD and high temperature (80-92°C). Thermophilic AD is suitable for POME due to low cooling cost requirements, and also increases in the rates of chemical and enzymatic reactions, an increase in thermodynamic favorability of methane production reactions [21]. However, the addition of OPA and biogas effluent recycling for pH adjustment of POME to improve the stability of AD reactor and biogas production under thermophilic conditions has never been studied. Proper amounts of oil palm ash addition and biogas effluent recycling into the AD reactors without disturbing the microbial communities or biochemical processes are required.
This work investigated the pH adjustment of POME by oil palm ash addition and effluent recycling as a pH regulator to improve the stability and feasibility of the process. The effects of pH adjustment of POME by oil palm ash addition and effluent recycling on change in dynamic bacteria and archaea population were investigated.”
- The experiment should be expanded, computer simulations added. This would allow other parameters of the experiment to be considered.
Response: I agree with the reviewer that computer simulations should give valuable information. However, computer simulations it not our expertise. The authors try to link the performance with the microorganism to allow the parameter for mean in terms of performance and response microorganisms inside the AD reactor. Section 3.4 page 12, line 314-351

Round 3
Reviewer 2 Report
The authors provided responses to all reviewer comments. I would only suggest changing the graphs to color, they will be more reader friendly. The conclusions of the manuscript could also have been highlighted more.
Author Response
Response to reviewers
Manuscript ID: processes-1196430
Type of manuscript: Article
Title: Enhancement of thermophilic biogas production from palm oil mill
effluent by pH adjustment and effluent recycling
The authors responded to all reviewer comments. I would only suggest changing the graphs to color. They will be more reader-friendly. The conclusions of the manuscript could also have been highlighted more.
Response: All graphs have changed to color. Figure 1 on page 4, Figure 2 on page 9, Figure 3 on page 11, and Figure 4 on page 12.
